# Ultrasensitive ELISA Developed for Diagnosis

**DOI:** 10.3390/diagnostics9030078

**Published:** 2019-07-18

**Authors:** Kanako Iha, Mikio Inada, Naoki Kawada, Kazunari Nakaishi, Satoshi Watabe, Yong Hong Tan, Chieh Shen, Liang-Yin Ke, Teruki Yoshimura, Etsuro Ito

**Affiliations:** 1Department of Biology, Waseda University, Tokyo 162-8480, Japan; 2R&D Headquarters, TAUNS Laboratories, Inc., Shizuoka 410-2325, Japan; 3Waseda Research Institute for Science and Engineering, Waseda University, Tokyo 162-8480, Japan; 4Department of Medical Laboratory Science and Biotechnology, Kaohsiung Medical University, Kaohsiung 80756, Taiwan; 5School of Pharmaceutical Sciences, Health Sciences University of Hokkaido, Hokkaido 061-0293, Japan; 6Graduate Institute of Medicine, School of Medicine, Kaohsiung Medical University, Kaohsiung 80756, Taiwan

**Keywords:** adiponectin, diagnosis, HIV, insulin, non-amplification nucleic acid detection, ultrasensitive ELISA

## Abstract

For the diagnosis of disease, the ability to quantitatively detect trace amounts of the causal proteins from bacteria/viruses as biomarkers in patient specimens is highly desirable. Here we introduce a simple, rapid, and colorimetric assay as a de novo, ultrasensitive detection method. This ultrasensitive assay consists of a sandwich enzyme-linked immunosorbent assay (ELISA) and thionicotinamide-adenine dinucleotide (thio-NAD) cycling, forming an ultrasensitive ELISA, in which the signal substrate (i.e., thio-NADH) accumulates in a triangular manner, and the accumulated thio-NADH is measured at its maximum absorption wavelength of 405 nm. We have successfully achieved a limit of detection of ca. 10^−18^ moles/assay for a target protein. As an example of infectious disease detection, HIV-1 p24 could be measured at 0.0065 IU/assay (i.e., 10^−18^ moles/assay), and as a marker for a lifestyle-related disease, adiponectin could be detected at 2.3 × 10^−19^ moles/assay. In particular, despite the long-held belief that the trace amounts of adiponectin in urine can only be detected using a radioisotope, our ultrasensitive ELISA was able to detect urinary adiponectin. This method is highly versatile because simply changing the antibody enables the detection of various proteins. This assay system requires only the measurement of absorbance, thus it requires equipment that is easily obtained by medical facilities, which facilitates diagnosis in hospitals and clinics. Moreover, we describe an expansion of our ultrasensitive ELISA to a non-amplification nucleic acid detection method for nucleic acids using hybridization. These de novo methods will enable simple, rapid, and accurate diagnosis.

## 1. Introduction

Examining patient specimens for the presence of proteins from pathogenic bacteria and viruses or as biomarkers enables rapid and accurate diagnosis, resulting in early treatment. Immunochromatography and enzyme-linked immunosorbent assay (ELISA) are commonly used methods for this purpose [1,2]. The sensitivity of these assays, however, requires further improvement. Although the use of radioisotopes enhances the sensitivity of these methods, preparing the special reagents and facilities for radioisotopes is difficult in hospitals and clinics. Another approach, mass spectrometry, is useful for identifying and measuring small quantities of proteins [3], but requires large and expensive equipment that may not be available at most medical facilities. More specifically, a user-friendly, simple, and rapid detection method for various marker proteins is needed for diagnosis.

The major challenge for protein quantification is that the proteins cannot be amplified like nucleic acids using polymerase chain reaction (PCR). We hypothesized that we could, however, amplify the detection signal of specific proteins. For the production of a user-friendly diagnosis system, we designed a method for amplifying the signals produced by the antigen-antibody reaction in ELISA. How can ELISA signals be amplified? One method is enzyme cycling [4]. One of the co-authors of the Kato et al. (1973) paper [4], Lowry, wrote in 1980, “Enzymatic cycling provides a methodology for virtually unlimited amplification of analytical sensitivity. The most widely applicable cycling systems are those for NAD and NADP, since these can be used to increase the sensitivity of methods for a host of other substances” [5]. Here NAD is nicotinamide-adenine dinucleotide. We thus considered that a combination of ELISA and enzyme cycling could be useful for early detection and herein describe our method for ultrasensitive ELISA.

The present review describes the detailed mechanisms of an ultrasensitive ELISA for diagnosis. This ultrasensitive ELISA consists of sandwich ELISA and thionicotinamide-adenine dinucleotide (thio-NAD) cycling. In the thio-NAD cycling, thio-NAD is reduced to thio-NADH, and the accumulated thio-NADH is measured at an absorbance of 405 nm using a microplate reader [6,7]. Examples of ultrasensitive detection of trace amounts of proteins for disease diagnosis have been described, including the detection of p24 for the diagnosis of an infectious disease (human immunodeficiency virus type 1: HIV-1) [8], and detection of adiponectin and insulin for diagnosis of lifestyle-related diseases [9,10,11]. We also introduce our attempt to expand the ultrasensitive ELISA to a non-amplification nucleic acid detection method, by combining nucleic acid hybridization and thio-NAD cycling.

## 2. Mechanisms of the Ultrasensitive ELISA

### 2.1. Principle of Protocol

The ultrasensitive ELISA that we developed consists of two parts: a sandwich ELISA and thio-NAD cycling (Figure 1) [6,7]. In the sandwich ELISA, the first antibody is used to immobilize the target protein to a microplate, and the second antibody is conjugated to an enzyme that converts the substrate to another form. Alkaline phosphatase (ALP, EC. 3.1.3.1) is used as this conjugated enzyme in our assay. A standard sandwich ELISA produces a color change in a substrate, resulting in a detectable signal. This signal increases in a linear fashion with time and thus the sensitivity is limited. Therefore, we considered that a substrate that can be hydrolyzed by ALP should be amplified. For this signal amplification, we adopted enzyme cycling. As described above, enzyme cycling is a method for amplifying a substrate [4]. It is generally expected that two kinds of enzymes are needed for the cycling reaction to act on the same substrate in different manners. In our version of enzyme cycling, however, we used a single enzyme, 3α-hydroxysteroid dehydrogenase (3α-HSD, EC. 1.1.1.50). The cofactors were NADH and thio-NAD, and we named our enzyme cycling ‘thio-NAD cycling’ [6,7].

The first substrate applied to ALP in the sandwich ELISA was 17β-methoxy-5β-androstan-3α-ol 3-phosphate. ALP hydrolyzes this substrate to 17β-methoxy-5β-androstan-3α-ol, which was not measured in our system. The 17β-methoxy-5β-androstan-3α-ol was used as the substrate for thio-NAD cycling. 3α-HSD with the cofactor thio-NAD oxidizes 17β-methoxy-5β-androstan-3α-ol to 17β-methoxy-5β-androstan-3-one, resulting in thio-NADH. Then, 3α-HSD with the cofactor NADH reduces 17β-methoxy-5β-androstan-3-one to 17β-methoxy-5β-androstan-3α-ol by the opposite reaction, resulting in NAD. Thio-NADH accumulates in a triangular fashion expressed as follows:(1)$$a\;\times \;b\;\times \;\overset{n}{\underset{k=1}{\sum }}k\;=\;a\;\times \;b\;\times \;\frac{n\left(n+1\right)}{2}$$
Here, *a* is the turnover ratio of ALP per min; *b* is the cycling ratio of 3α-HSD per min; and *n* = minutes of measurement time. The amount of the target protein is determined by measuring the absorbance using a microplate reader at 405 nm, which corresponds to the maximum absorption wavelength of thio-NADH (exactly 400 nm). Therefore, the signals to be used for quantifying the proteins can be obtained in a short period of time.

### 2.2. Experimental Protocol

We show the typical protocol to detect a reference ‘adiponectin’ antigen using ultrasensitive ELISA as follows. This protocol was modified from Watabe et al. [6].

1. Coat a primary antibody.

Dilute the first antibody with 50 mM Na_2_CO_3_ (pH 9.6) to a concentration of 2 μg/mL. Add 100 μL of the antibody into each well of 96-well microplates. Incubate for 1 h at room temperature.

2. Wash microplates.

Wash the microplates 3 times with TBS including 0.05% Tween 20.

3. Block nonspecific binding sites.

Dilute the TBS including 10% BSA 10 times with TBS. Block nonspecific binding sites by filling wells with this solution at 300 μL/well. Incubate for 1 h at room temperature.

4. Wash microplates.

Wash the microplates 3 times with TBS including 0.05% Tween 20.

5. Add an antigen.

Dilute the antigen with TBS including 0.1% BSA to 5–40 pg/mL. Add 100 µL of this antigen solution to each well. Incubate overnight at 4 °C.

6. Wash microplates.

Wash the microplates 9 times with TBS including 0.05% Tween 20.

7. Add an enzyme-linked secondary antibody.

Dilute an enzyme-linked secondary antibody with TBS including 0.1% BSA and 0.02% Tween 20 to 10 pmol/mL. Add 100 µL of this antibody solution to each well. Shake the microplates for 1 h at room temperature.

8. Wash microplates.

Wash the microplates 9 times with TBS including 0.05% Tween 20.

9. Add a thio-NAD cycling solution.

Dissolve 1 mM NADH, 3 mM thio-NAD, 0.1 mM 17β-methoxy-5β-androstan-3α-ol 3-phosphate, and 30 U/mL 3α-HSD into 0.1 M Tris-HCl (pH 9.5). This is referred to as a thio-NAD cycling solution. Add 100 µL of this thio-NAD cycling solution to each well.

10. Measure absorbance.

Measure the absorbance at 405 nm and 660 nm with a microplate reader every 5 min for 1 h at 37 °C. The absorbance at 660 nm is used as a reference for background correction.

For detection of the specimens obtained from patients (i.e., serum, urine, and saliva), we always checked a spike-and-recovery test. The spike-and-recovery test is a technique for analyzing and accessing the accuracy of ELISA for particular specimen types. It is used to determine whether analyte detection can be affected by the difference between the diluent used for preparation and the experimental specimen matrix. To perform a spike-and-recovery test, a known amount of analyte (i.e., target protein) is added to a matrix (i.e., specimens). This ‘addition’ is called a ‘spike’. The concentration of the added analyte in the matrix is determined from standard curves prepared. The concentrations denote the spike recovered in the matrix. All the experimental procedures for this test were the same as for the above ultrasensitive ELISA experiments without the antigen solution.

We determined the limit of detection (LOD), the minimum limit of quantification (LOQ), and coefficient of variation (CV) for the following sections. The experimental data were obtained by subtracting the mean value of the blank signals from each of the corresponding measured data points. The LOD was estimated from the mean of the blank, the standard deviation of the blank, and a confidence factor of 3. The minimum LOQ was estimated by the same method that was used for the limit of detection, but with a confidence factor of 10. The CV calculated from 3 data points was obtained for a reference antigen at a given concentration, and the value should be under 10%.

## 3. Detection of Trace Amount of Proteins

### 3.1. Detection of Proteins for an Infectious Disease

As an example of a diagnosis of an infectious disease, we applied our ultrasensitive ELISA to detect a specific protein for diagnosing HIV infection [8]. The fourth- and fifth-generation HIV tests need to detect the HIV-1 and HIV-2 antibodies and the HIV-1 p24 antigen [12]. Thus, we focused on the detection of HIV-1 p24 protein. For a CE-marked HIV antigen/antibody assay, the LOD of HIV-1 p24 must be less than 2 IU/mL [13,14]. A recent study revealed that the LOD of HIV-1 p24 among various FDA-approved HIV antigen/antibody combination tests ranges from 0.19 to 1.77 IU/mL [15].

The LOD and minimum LOQ for HIV-1 p24 in our ultrasensitive ELISA was 0.0065 and 0.0242 IU/assay (i.e., ca. 10^−18^ and 10^−17^ moles/assay), respectively [8]. Because our assay contained 50 μL of solution, the LOD value is expressed as ‘per mL’ (0.13 IU/mL). Our ELISA system included a washout process, therefore we consider that the absolute value, and not the concentration, is important (i.e., 0.0065 IU). The result of our ultrasensitive ELISA was thus better than those of the FDA-approved tests, indicating that our ultrasensitive ELISA for HIV-1 p24 can reduce the ‘window period’ required for the diagnosis of HIV by detecting the protein earlier in the infection process.

In our previous study, we compared the LOD between our ultrasensitive ELISA and a nucleic acid test (NAT) [8]. The number of HIV-1 p24 proteins in the virion is thought to be larger than the number of RNA copies (approximately 3000 HIV-1 p24 proteins vs. 2 RNA copies per virion) [16]. The value of 10^−18^ moles is almost the same as 10^6^ proteins, corresponding to ca. 10^3^ RNA copies. The NAT (i.e., real-time PCR) can detect 10^1^- or 10^2^-order copies of nucleic acids [17]. That is, the LOD of our ultrasensitive ELISA is approaching the LOD of the NAT with a margin of only one or two orders of magnitude. In addition, spike-and-recovery tests using blood confirmed the reliability of our ultrasensitive ELISA [8].

### 3.2. Detection of Proteins for a Lifestyle-Related Disease

Adiponectin is an adipocyte-derived vasoactive peptide [18]. Serum adiponectin enhances insulin sensitivity, and individuals with obesity, type 2 diabetes mellitus (DM), and other metabolic disorders have low serum adiponectin levels [19]. On the other hand, urinary adiponectin is a useful marker of the progression of diabetic nephropathy, indicating that DM patients have high urinary adiponectin levels [20]. The sensitivity of commercially available ELISA kits, however, is insufficient to measure the low level of urinary adiponectin in normal subjects [21,22]. We thus attempted to apply our ultrasensitive ELISA to detect urinary adiponectin levels in normal subjects and distinguish between normal subjects and DM patients [10,11].

The LOD of adiponectin and minimum LOQ were 0.81 pg/mL (i.e., ca. 2.7 × 10^−19^ moles/assay) and 2.7 pg/mL (i.e., ca. 9.0 × 10^−19^ moles/assay), respectively, for our ultrasensitive ELISA, when the molecular mass was assumed to be 300 kDa and the volume of the assay was 100 μL. Our ultrasensitive ELISA, therefore, succeeded in detecting urinary adiponectin at the subattomole level. We then attempted to determine the urinary adiponectin concentrations in healthy subjects and DM patients. The urinary adiponectin concentrations were corrected on the basis of the creatinine concentrations. The mean urinary adiponectin levels of healthy subjects were 3.06 ± 0.33 (ng/mg creatinine, mean ± SEM), and those of the DM patients were 14.88 ± 3.16 (ng/mg creatinine). That is, the urinary adiponectin levels were significantly higher in DM patients (*p* < 0.05) than in healthy subjects. Further, a threshold of urinary adiponectin levels could be set at 4.0 ng/mg creatinine to distinguish between healthy subjects and DM patients [11].

In 2019, our further progress showed that the urinary adiponectin levels increased with an increase in the chronic kidney disease risk and that urinary adiponectin mainly formed a medium-molecular weight multimer in DM patients, whereas it formed only a low-molecular-weight multimer in healthy subjects [11]. That is, we concluded that urinary adiponectin can become a new diagnostic index for chronic kidney disease. Our assay is a noninvasive test using only urine, thus reducing the patient burden.

The second target protein for a lifestyle-related disease was insulin [6,9]. An ELISA for insulin was established in the 1980s and 1990s, and the LOD was on the order of μIU/mL, corresponding to tens of femtomoles/mL. The molecular weight of human insulin is 5807, and for this conversion 1 IU was estimated as ca. 43 μg [23]. We needed to exceed this sensitivity for our test.

We were faced with a fundamental problem for detecting insulin. Although many hospitals and diagnostic companies use them, the reference material of insulin, i.e., the quality of the World Health Organization (WHO) international standard insulin reference or its equivalent products, is poor. As a practical manner, the use of recombinant human insulin is strongly advised from the standpoint of calibration traceability [24]. When the WHO international standard insulin reference (see [22]) was applied to our ultrasensitive ELISA, the LOD was 19 nIU/assay, corresponding to ca. 1.4 × 10^−16^ moles/assay [9]. These values are 0.38 μIU/mL and ca. 2.8 × 10^−15^ moles/mL for a 50-μL assay volume, and thus they were somewhat better than the previously reported values.

When we used a recombinant insulin reference (MP Biomedicals, MP Bio Japan, Tokyo, Japan) in our ultrasensitive ELISA, the LOD was 0.0047 pg/assay (i.e., 8.0 × 10^−19^ moles/assay) [6]. Because the assay volume was 50 μL, we could detect insulin in the order of tens of attomoles/mL. That is, the LOD of our assay was at least three orders of magnitude more sensitive than those of the previously reported assays.

Ultrasensitive detection of proteins involved in lifestyle-related diseases is important. Patients with these diseases, such as DM, must continuously provide blood samples for repeated testing. If the LOD of a specific protein can be improved for early diagnosis, it may obviate the need for blood sampling. Urine, saliva, and tears may include the same proteins, but in very small amounts when compared to blood. Thus, our ultrasensitive ELISA can be used for the noninvasive detection of proteins in urine, saliva, and tears, thereby alleviating patient pain and discomfort.

We are now applying this ultrasensitive ELISA to detect specific proteins from active bacilli [2] and pathogenic viruses as well as proteins that can be used as cancer markers.

## 4. Challenges: Expansion of the Ultrasensitive ELISA to a Non-Amplification Nucleic Acid Detection Method Using Hybridization and Thio-NAD Cycling

The value of 10^−18^ moles in the protein experiments is equivalent to approximately 10^6^ molecules. We hypothesized that if we replaced the antigen-antibody reaction for proteins with the hybridization of nucleic acids, nucleic acid detection could be performed at the level of 10^6^ copies in another new assay. In the planned system, we will prepare two different nucleic acid probes, similar to the two different antibodies in ELISA, one of which is used to immobilize the target nucleic acid sequence and the other for ALP labeling. Here, we should consider the size of ALP compared with that of the nucleic acid probes. The molecular mass of ALP is 80,000–150,000 Da. As expected, our preliminary experiments indicated that locating the ALP near the probes inhibits hybridization, resulting in poor sensitivity. We thus consider that a small spacer is needed between the probes and the ALP. We are now thinking that fluorescein isothiocyanate ((FITC), molecular mass 389 Da) could function as this spacer. Thus, the secondary probe is linked to FITC, and the FITC binds to an anti-FITC antibody that is labeled with ALP. Our preliminary data suggested that we could measure nucleic acids at 10^6^ copies/assay. Further, by increasing the number of secondary probes, we expect to improve the sensitivity.

Why are we working to develop a new nucleic acid detection system with a sufficiently high sensitivity to replace PCR? We think that PCR has many critical drawbacks, including, e.g., (1) non-specific or false-positive amplifications, (2) target sample volume limits, (3) deactivation of amplification enzymes, (4) the complicated techniques, (5) the difficulty in designing probe sequences, and (6) expense. Non-specific or false-positive amplifications occur due to excess DNA input, long targets, or contamination. The sample amount used is ca. 1 μL, which means that at least 1000 copies must be included in a 1-mL volume. This low concentration may contribute to the production of false negative results. The deactivation of enzymes further deteriorates the amplification efficiency. Further, the PCR techniques are complicated. A new method that can handle a large volume and avoid amplifying the nucleic acids is therefore needed. We refer to our new nucleic acid detection system as a ‘non-amplification nucleic acid detection method’. This assay can be used not only in laboratories but also on-site in place of PCR.

## 5. Conclusions

We developed an ultrasensitive ELISA comprising of a sandwich ELISA and thio-NAD cycling to detect trace amounts of proteins for the diagnosis of disease. This system is also suitable for single-cell analysis [7]. Expanding this ultrasensitive ELISA to develop an assay for nucleic acids will overcome the drawbacks of PCR, and this non-amplification nucleic acid detection method is potentially also widely applicable for diagnosis.

## Figures and Tables

**Figure 1 diagnostics-09-00078-f001:**
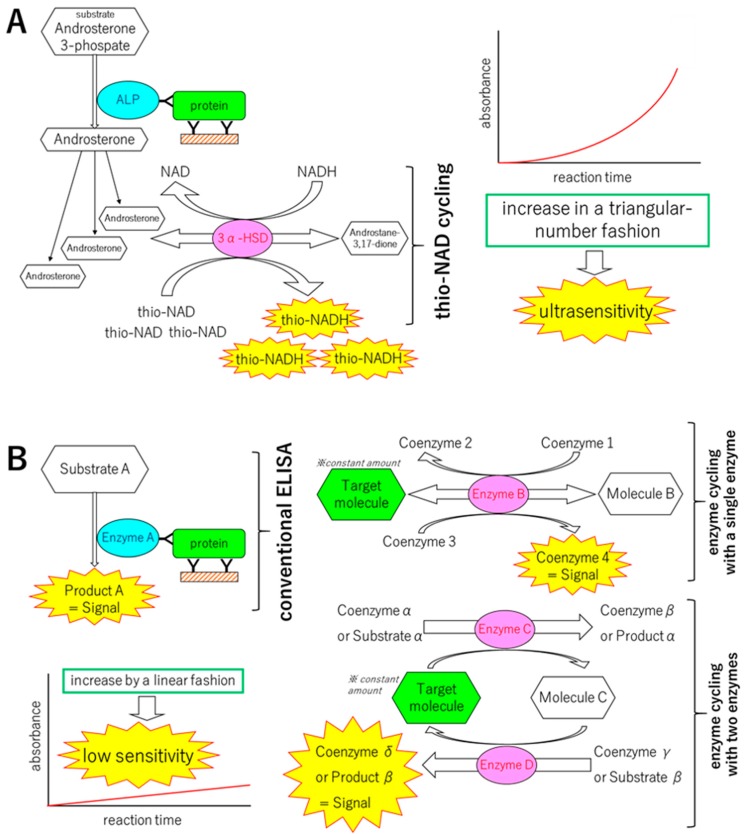
Schematics of ultrasensitive ELISA, standard ELISA, and standard enzyme cycling. (**A**) An ultrasensitive ELISA consisting of a sandwich ELISA combined with thionicotinamide-adenine dinucleotide (thio-NAD) cycling. Two antibodies used in ELISA specifically target a pathogenic protein. The first antibody is used for immobilization, whereas the second antibody is labeled with alkaline phosphatase (ALP), which hydrolyzes a substrate containing phosphate. The hydrolyzed substrates are used in thio-NAD cycling that employs a main enzyme (dehydrogenase) and its coenzymes (NADH and thio-NAD). Thio-NADH accumulates in a triangular manner and can be measured at 405 nm. 3α-HSD is 3α-hydroxysteroid dehydrogenase. (**B**) A standard, conventional ELISA and two kinds of standard enzyme cycling (a single enzyme system and a two-enzyme system). The standard ELISA, the two kinds of enzyme cycling and even the sequential performance of a standard ELISA followed by an enzyme cycling show the signals in a linear function and the resulting low sensitivity, whereas our ultrasensitive ELISA has higher sensitivity than these standard methods. *The target molecule is not increased during the reactions.

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
