# Peer review of "Ultrasensitive ELISA Developed for Diagnosis"

_diagnostics, 2019, doi:10.3390/diagnostics9030078_

Reviewer 1 Report

This is a rather well writer review of a potentially very interesting highly sensitive approach on which the authors have invested a considerable amount of effort. Their plans to expand it to nucleic acids and replace PCR are also intriguing and their communication to interested researchers may also prove useful for the exploitation of this idea.

However, I think the manuscript is missing a few graphs which would compare the ultrasensitive ELISAs with the current assays for the same proteins, so as the reader can easily realize the advantages of the new technique.

Author Response

As suggested by the Reviewer 1, we added a new figure to compare our ultrasensitive ELISA with the current assays (see Figure 1B and line 104). Then, the comparison of data between an ultrasensitive ELISA and the current methods has been already shown about p24 in the lines of 176 - 180, and we have also claimed that urine adiponectin cannot be determined using the current methods but did by an ultrasensitive ELISA (see lines 196 - 204).

Reviewer 2 Report

The manuscript is well written and can be published

Author Response

Thank you.

Reviewer 3 Report

The paper is well written and resume an intresting methods and some of its application, however, the paper was submitted as a Review. The aim of a review according to instruction for authors is: provide concise and precise updates on the latest progress made in a given area of research".

As all result presented in the present paper have been already published between 2014 and 2016 I do not see any news in the paper. 

Author Response

We added the new data published in 2019. It showed that when only our ultrasensitive ELISA is used, urinary adiponectin can become a new diagnostic index for chronic kidney disease due to diabetic nephropathy. Please see lines 213 - 218. Further, the language has been already edited before submitting our manuscript by an American scientist.

Round  2

Reviewer 3 Report

I understand the response of the authors and I appreciate the adding of new data. It is my personal opinion that maybe the paper could be focused only on adiponectin and not presented as a review.